# Research on measurement indexes and evaluation for the collaborative efficiency of emergency information sharing in coastal cities of China

Zou Jiang[1], Huang Xing [2]*

**1** School of Economics and Management, Mianyang City College, Mianyang, China, **2** School of Economics and Management, Southwest University of Science and Technology, Mianyang, China

\* huangxing6213@126.com

## Abstract

### Background

The paper established indexes of coordination efficiency, the theoretical framework and operation mechanism of emergency information sharing for coastal cities of China.

### Method

First of all, we analyzed the operational relationship between the participants of emergency information sharing and information transmission, and based on the collaborative theory, constructed the emergency information sharing framework and operational mechanism suitable for the actual disaster prevention and reduction of coastal cities. Around the 3 dimensions of emergency information sharing mechanism construction, resource guarantee ability and collaborative driving force of emergency information sharing, the paper proposed the evaluation index system and the evaluation method.

### Results

The empirical results showed that the efficiency of emergency information sharing in coastal cities in China was generally low, and the contribution rate of the construction level of emergency information sharing mechanism is higher than that of the resource guarantee ability and the collaborative driving force of emergency information sharing, but the efficiency of emergency information sharing in coastal cities in China was still at the bottom level.

### Conclusion

The research results provided theoretical basis and methods for the emergency management departments of coastal cities in China.

**Data Availability Statement:** All relevant data are within the paper.

**Funding:** This article is funded by the National Natural Science Foundation of China (72072020),

China Central University Project(2021SYB10), Key Project of Sichuan Circular Economy Research Center (4XHJJ-2102, XHJJ-2101) and Key Project of Sichuan Information Management and Service Research Center (SCXX2020ZD02). The funders had no role in study design, data collection and analysis, decision to publish, or preparation of the manuscript.

**Competing interests:** The authors have declared that no competing interests exist.

## Introduction

China's coastal cities have a large population and rich marine resources. In recent years, natural disasters have occurred frequently due to many factors such as special geographical locations and complex weather systems, which have greatly affected the social and economic stability and development of China's coastal cities [1]. At present, an emergency management department has been established to solve the problems of fragmentation and low coordination, which can greatly improve the coordination ability of coastal cities in disaster prevention and mitigation. Overall, China's coastal cities have achieved great improvement in their ability to coordinate disaster prevention and mitigation, and a comprehensive coordination and joint prevention and control mechanism has been basically formed. Especially in the process of disaster prevention and reduction, some problems include untimely information sharing, unclear information tasks and unsatisfactory operation mechanism of confidence sharing, etc., which have greatly restricted the efficiency and effectiveness of disaster prevention and reduction in coastal cities of China. Therefore, how to build a scientific and reasonable emergency information sharing mechanism for disaster prevention and mitigation in coastal cities, and how to improve and innovate the traditional "command-control" emergency information supervision mode have become some urgent problems to be solved.

At present, emergency information sharing is a hotspot problem in emergency management. Scholars have carried out in-depth research on emergency information sharing collaborative, emergency information sharing mechanism and emergency collaborative governance. Research on emergency information sharing and collaboration mainly focuses on emergency information collaborative factors, collaborative framework and collaborative efficiency. Chen Yumei (2018) [1] believed that the five aspects of support within the organization, coordination between organizations, legal protection and supervision incentives and external environment are the key factors; Zhang Xinrui et al. (2022) [2] analyzed the function relationship between the information collaborative elements, distinguished the cause element and the result element; Based on the core blockchain technology distributed ledger technology, asymmetric encryption algorithms and smart contracts, Yin Yong (2021) [3] combined the needs of smart medical multi-information management, constructed health information alliance chain, personal health information data private chain, public opinion charity information public chain under public health emergencies; Zhang Guirong et al. (2022) [4], analyzed the subject, content, form and application of emergency information collaboration and constructed an emergency information with the power mechanism, operation mechanism and action mechanism as the core collaboration mechanism; Based on the characteristics of blockchain technology, Jiang Yun et al. (2021) [5] established an internal private chain, organizational alliance chain, and social public chain architecture according to the needs of emergency information resources, and analyzed the factors that improve the efficiency of information collaboration; Delone et al. (1992) [6] summarized the six dimensions of information system success: system quality, information quality, system use, personal impact, user satisfaction, and organizational impact, etc. At the research of emergency information sharing mechanism, the most studied are emergency information sharing decision-making and operation mechanism.

The research on the emergency information sharing mechanism mainly focus on the emergency information sharing decision-making and operation mechanism. Chen Yumei et al. (2017) [7] put forward policy recommendations to improve the efficiency of emergency information sharing around the legal system, shared concepts, information technology and supervision and incentives; based on the theory of tripartite evolutionary game, Zheng Wanbo et al. (2022) [8] used computer software to simulate the evolution process and results of behavior strategies of each participant under different constraints, and discussed the corresponding

optimal information sharing strategy; Carminati et al. (2013) [9] provided a decision-making system for implementing timely and controlled information sharing in emergency situations; Zhang Ping et al. (2013) [10] analyzed and discussed the obstacles and corresponding counter-measures of emergency management information sharing. In terms of research on the operation mechanism of emergency information sharing, Meng Qingping et al. (2011) [11] designed and implemented a large-scale urban district-level integrated management system, and analyzed the overall structure, software functions, information sharing mechanism and business collaborative model; Zeng Yuhang et al. (2012) [12] started from the analysis of emergency management information processing process, and proposed an emergency information coordination mechanism model for the reasons for the difficulty of information coordination in emergency information management, and discussed the operation of emergency information coordination and sharing mechanism in the e-government environment; Du Jun et al. (2017) [13] analyzed the characteristics and specific paths of emergency information transmission to base on the organizational framework of emergency network, and finally studied the mechanism of emergency information sharing and the network model of emergency information sharing; Carminati B et al. (2016) [14] presented an emergency information sharing framework able to deal with both specified and unspecified emergencies; Dantas A A et al. (2007) [15] presented an information-sharing framework for road organizations, on the basis of a study of response and recovery activities, information needs were identified and a geographic information system-based information-sharing framework was created. In the research of emergency collaborative governance, Wang Ying et al. (2016) [16] conducted a SWOT analysis of multi-subject collaboration in urban emergency management by introducing the theoretical framework of collaborative governance, in order to realize the innovation of urban emergency management model; Based on the perspective of resource allocation system analysis, Liao Chuhui (2020) [17] used collaboration theory and binary nonlinear method to study the mechanism, implementation path, specific scheme of cross-departmental resource information integration and collaborative governance in public emergencies; Wang Yu (2020) [18] analyzed the problem of incompatibility between the coupling characteristics of environmental emergencies and the traditional emergency management model based on collaborative governance theory; Lin Zhen (2019) [19] studied the collaborative governance mechanism of online public opinion for emergencies composed of data, tools and business process reorganization, forming a multi-agent system governance.

The above literature mainly studies the management of emergency information sharing from management, although there have been many discussions on the synergy of emergency information sharing and its influencing factors, its organizational structure and its efficiency, but these results are more focused on the discussion of macro-decisions, and their shortcomings are reflected in: First, the existing achievements are lack of a set of systematic emergency information sharing mechanism from the micro perspective. The description of the supply and demand subjects of emergency information resources is not clear; Second, the mechanism of emergency information sharing in coastal cities has its own characteristics. Most of the information resources come from marine disasters. It is necessary to build a set of emergency information sharing mechanism in line with the specific disasters in coastal cities of China. The existing research is lack of this aspect; The third is the lack of a set of measurement index systems and methods that can effectively evaluate the efficiency of emergency information sharing in coastal cities for disaster prevention and mitigation, making the emergency information sharing operation mechanism lacking evaluation basis. In view of this, this paper aims at the characteristics of coastal city disaster prevention and mitigation, from the perspective of synergy theory, by clarifying the relationship between the participants of emergency information sharing and information transmission, constructs an emergency information sharing

framework and operation mechanism suitable for coastal city disaster prevention and reduction. On this basis, the measurement index system and evaluation method of emergency information sharing collaborative efficiency are further proposed, with a view to providing theoretical guidance and method support for the practice of disaster prevention and mitigation information management in coastal cities. The relationship between this paper and the previous research lies in: First, the index system of the existing research is optimized, and an effective evaluation index system is put forward from three dimensions, included the mechanism construction level of emergency information sharing, the resource support capability and the collaborative driving force. Secondly, the coupling theory in physics is used for reference to quantitatively evaluate the coordination efficiency of emergency information sharing in coastal cities. Thirdly, referring to the existing emergency information organization structure, the paper puts forward the emergency information sharing framework of China's coastal cities based on collaboration theory. The difference of this paper is that it proposes a coastal city emergency information sharing framework and operation mechanism based on the synergy theory, and proposes an index system and measurement method that affect the coastal city emergency information sharing collaboration efficiency.

## Emergency information sharing framework based on collaboration theory

### Connotation of collaboration theory

Collaboration theory was put forward by German physicist Herman Haken in 1976. Collaboration theory mainly studies how open systems far from equilibrium can spontaneously appear orderly structures in time, space and function through their own internal synergism when they have material or energy exchange with the outside world. The system is regarded as a complex open system composed of three elements: human, organization and environment in the collaboration theory. Each element is nested with multiple sub-elements, and its interior presents nonlinear characteristics. Although different subsystems have different attributes, there is a cooperative relationship between each subsystem. Whether the system can play a synergistic benefit is determined by the synergy of the subsystems within the system. To promote the synergy of each subsystem, the system must be open and able to communicate with the outside world. The exchange of energy and information ensures that the system is capable of development. Collaboration theory deals with complex systems from the perspective of systems view, and provides a theory and method for dealing with decentralized and disordered emergency management systems, making it a system of virtuous circles of coordination, cooperation and order.

### Elements of the emergency information sharing framework of coastal cities

This paper builds a coastal city emergency information sharing framework based on collaboration theory. The coastal city emergency information sharing framework consists of four elements: emergency information participants, emergency information resources, emergency behavior, and emergency resource software and hardware resources. According to the collaborative rules, it can form an organic whole with the characteristics of subject coordination, sharing efficiency, and rapid response, which can maximize emergency efficiency and benefits. The relationship between the components of the emergency information sharing framework is interrelated and interdependent: emergency information participants and emergency information resources provide the basis for emergency behaviors, and emergency behaviors are supported by emergency software and hardware resources to achieve the goals of emergency

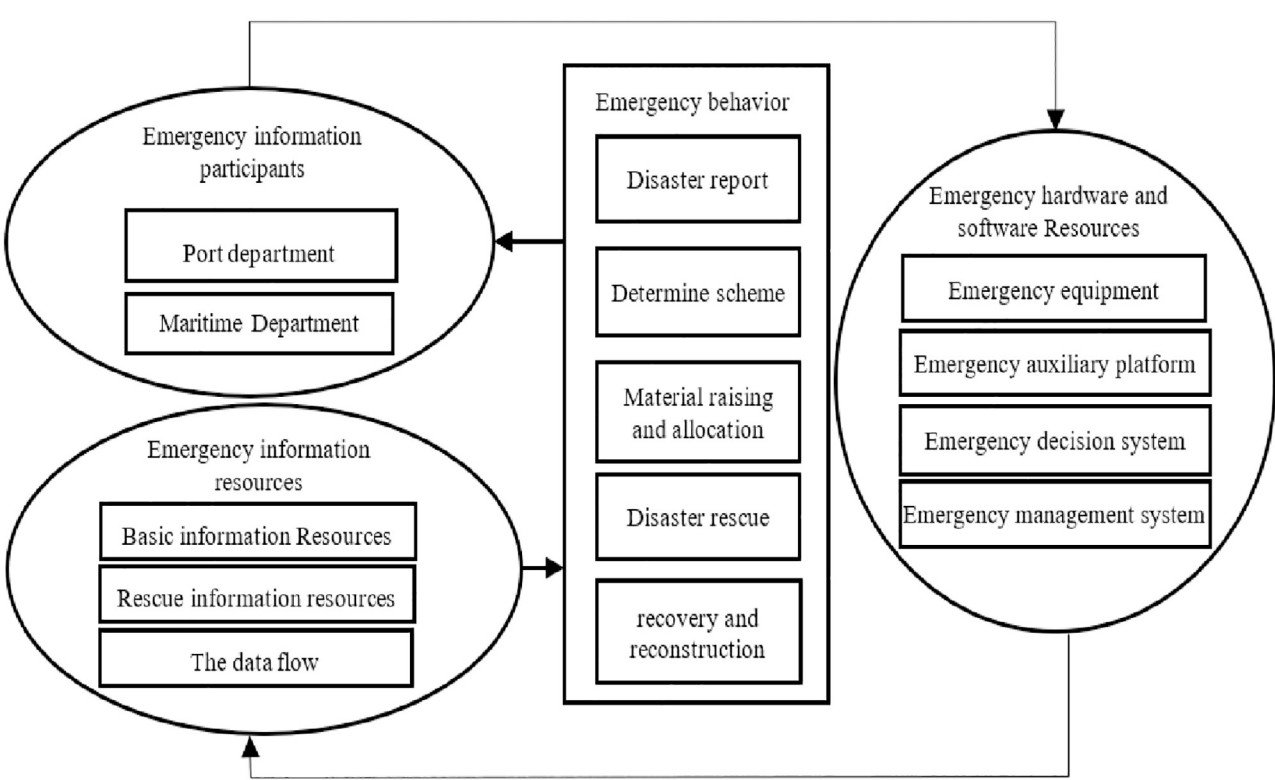

**Fig 1. The relationship between emergency information participants, emergency information resources, emergency behaviors, and hardware and software.**

information participants and emergency resources (Fig 1). The participants of coastal city emergency information are mainly composed of people and organizations that provide, process, analyze and use information, including: government, port and shipping, maritime, public security and fire protection, enterprises, social organizations, information management, medical care and other related institutions, as well as expert groups, rescue teams, the public, the military, media workers, technicians and other related personnel; Emergency information resources are composed of basic information resources, rescue information resources, and data streams. Basic information resources include five aspects: politics, economy, social development, resources and environment, and geographic information. Emergency information resources include disaster information, emergency plans, and emergency response. (Kou Youguan, 2005) [20]; Emergency behaviors include disaster reporting, determination of plans, material financing and deployment, disaster relief and recovery, etc.; hardware and software resources include facilities and equipment for storage, reception, processing, and transmission of emergency information, and decision-making systems and assistance to support emergency operations Platforms and management systems, etc. (Xiao Hua et al., 2015) [21].

## Coastal city emergency information sharing framework based on synergy theory [22–25]

At present, China's emergency response mode mainly adopts the two-level mode of emergency command center and emergency response department. The emergency command center is mainly responsible for emergency decision-making and command and coordination

functions. It belongs to the coping mode of centralized management and decentralized implementation.

This two-level response model can effectively ensure the efficiency of information decision-making and the unified command of decision-making, but there are also many problems in the process of emergency information sharing. One is the smoothness of information communication channels and communication efficiency. It is prone to miscommunication between the emergency subject and the public, resulting in deviations in the information received by the emergency subject, and the emergency command agency has unclear needs for public information; the second is the problem of departmental coordination. It is not conducive to the coordination of the collection, processing and feedback mechanism of emergency information between departments; third, the emergency response department lacks the initiative of the main body. and feedback, etc. are inefficient. The idea of the synergy theory lies in the diversification of the main body and the synergy of the cooperation methods. Its purpose is to maximize the value of resource utilization and service quality by integrating multiple resources, which provides a theoretical basis for the synergy of emergency information sharing in coastal cities in China. In Fig 1 above, the three elements of emergency information participants, emergency information resources, and emergency software and hardware resources ultimately provide support for emergency behaviors, and achieve disaster prevention and mitigation goals through emergency behavior. However, the execution effect of emergency behaviors is affected by emergency information participants, emergency information resources, and hardware and software. Emergency information participants are the performers and decision makers of emergency behaviors.

The synergy of the two will affect the efficiency of emergency behaviors, and the efficiency of emergency behaviors is affected by the degree of information sharing, information coverage, information processing capabilities and decision effectiveness, only when they cooperate with each other and respond in a coordinated manner can the level of emergency information sharing be promoted. Therefore, the prerequisite for improving the level of emergency information sharing in coastal cities is to build a coordinated and effective information resource sharing network. Such a network should be composed of the main body of emergency response information generation, management and use, and information transmission channels, management platforms, software and hardware, etc. Co-participate in a shared collaborative system with networked, integrated, and digital features to achieve cross-platform, cross-department, and cross-regional information collaboration services. This article is based on the synergy theory and on the basis of reference four, proposes the emergency information sharing framework shown in Fig 2.

Fig 2 improves the traditional response model of centralized management and decentralized implementation into an emergency information sharing model of centralized management, decentralized implementation, subject coordination, and joint participation. This new sharing model is beneficial to make up for the information islands, The lack of a single source of information and the inadequacy of the information supervision mechanism have effectively realized the co-construction and sharing and efficient collaboration of disaster prevention and mitigation emergency information in coastal cities.

**Operating mechanism of emergency information sharing framework from a collaborative perspective.** The main body of emergency information generation and use of emergency information in Fig 2 includes government agencies, non-governmental organizations, enterprises and institutions and expert groups, the public, rescue teams, media and other related organizations and individuals. The emergency information management center is the core organization for sharing emergency information in coastal cities. It is mainly responsible for the formulation of emergency information disposal standards, emergency information

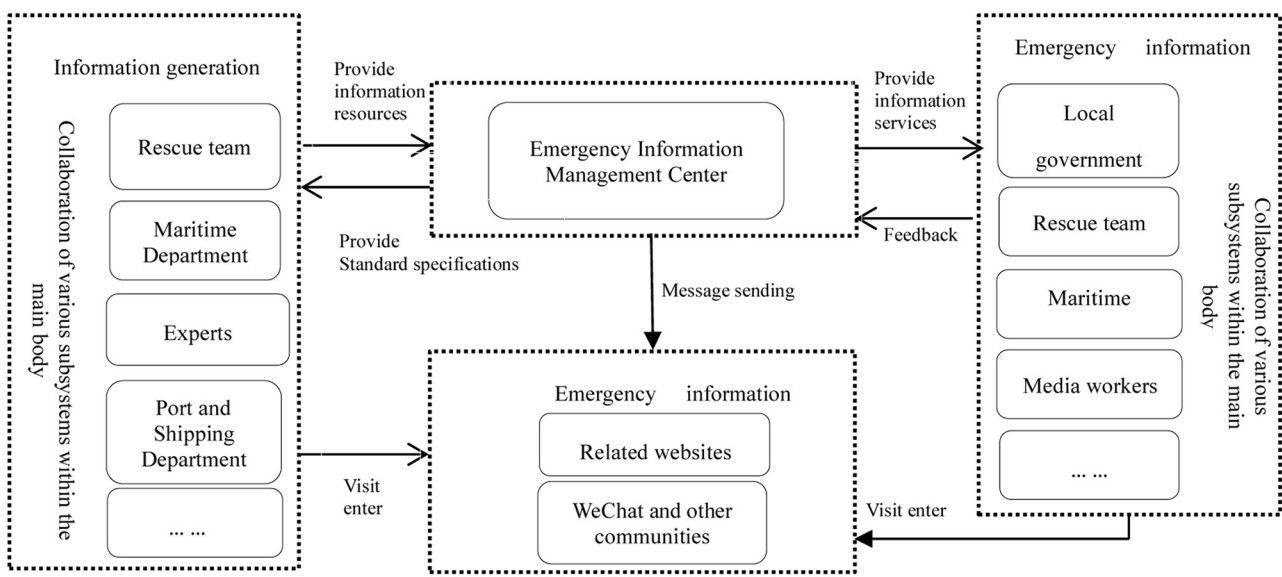

**Fig 2. Coastal city disaster prevention and mitigation emergency information sharing framework based on collaboration theory.**

services, distribution and comprehensive management. The emergency information management center will provide the processed emergency information resources to the emergency information use subject, and the emergency information use subject will feed back the information to the emergency information management center according to the decision-making needs and execution needs. The available shared information is formed, and the emergency information management center sends the shared information to the media in a unified manner for all subjects and people to access. The entire operation process not only embodies centralized management and decentralized implementation, but also embodies the coordination and joint participation of the subjects.

**Establish a coordinated and standardized platform for emergency information sharing.** The function of the emergency information sharing collaborative standardization platform is to synthesize the information of different departments, different regional platforms and various types of users, and make emergency decisions based on the recommendations of expert groups. Through the emergency information sharing collaborative platform, users can conduct a one-stop search and can support each coastal city to provide emergency information sharing services to the user body. The standardized platform for emergency information sharing mainly integrates various local governments, maritime departments, port and shipping departments, public security and fire protection, enterprises, non-social organizations and other emergency information participants in coastal cities. Standardize information resources and information services through a standardized platform, including information resource management standards, information security technology standards, information conversion standards, system integration standards, system operation standards, and information service standards, etc. The purpose is to break the pattern of individualism.

**Establish a collaborative communication model for emergency information.** The collaborative communication of emergency information involves the entire emergency management area of the coastal city. It is different from the traditional single-media independent communication mode. It requires the close cooperation and joint efforts of the relevant departments of the entire coastal city. In terms of establishing inter-departmental coordination

and cooperation, accomplish tasks that cannot be completed by a single coastal city. In order to establish an emergency information collaborative dissemination model, it is necessary to reform the traditional emergency information dissemination model, break the boundaries between departments, enter emergency information into meta data according to the sharing standard, and establish shared links between departments and websites. By establishing an emergency information collaborative communication model, the sharing of emergency information resources in coastal cities will be gradually realized.

**Establish a collaborative management model for emergency information resources.** The collaborative management mode of emergency information resources based on the collaboration theory mainly starts from establishing the emergency knowledge element link mode, and describes emergency information by means of knowledge elements. The coordination of emergency information resources can be reflected as the content coordination between emergency knowledge elements; Then, the constantly updated, massive, and constantly emerged emergency knowledge elements are built into the emergency knowledge meta-database, and then the emergency knowledge meta-database is embedded into various databases. Through this process, the emergencies knowledge database can embed the database and the emergency information between the platforms is related according to the content, forming an organic whole. On this basis, a hierarchical and modular approach is used to divide the emergency information resources into a multi-knowledge element and multi-level emergency information resource management system to realize the cross-domain of the emergency information resource collaborative management model based on the emergency knowledge element.

## Coastal city emergency information sharing collaborative efficiency measurement

### Measurement indexes of emergency collaborative efficiency of emergency information sharing

This paper proposed the basis for measuring the efficiency indexes of coastal city emergency information sharing collaboration as follows:

**Based on existing research results.** The research results of Hu Ping (2007) [26], Sharon S (1996) [27] and David L (2001) [28] were used in the proposed index for measuring the efficiency of emergency information sharing collaboration. These indexes include 3 primary indexes and 9 secondary indexes. The first fist-level index is the construction level of emergency information sharing mechanism, which has four secondary indexes, including emergency information reporting, emergency information security, and emergency information sharing incentives. The second first-level index is the resource support capability of emergency information sharing, which has two second-level indexes, including the unity of emergency information standards, and financial support. The third first-level index is the collaborative driving force of emergency information sharing, and it has three second-level indexes, including clear management functions, obstacles to emergency information sharing, and the integration ability of emergency management agencies. These indexes can scientifically evaluate the collaborative efficiency of information sharing in coastal cities.

**Based on in-depth interviews with coastal city emergency management departments, port and shipping departments, and maritime departments.** Through in-depth interviews with relevant departments in 6 coastal cities, including the Haikou Municipal Port and Shipping Bureau, the Shanghai Emergency Management Bureau, and the Fuzhou Maritime Safety Bureau, the commonality of emergency information sharing efficiency measurement indicators was summarized.

**Principles of index selection.**

1. *Systematization principle*. Indexes can reflect both direct and indirect effects, and ensure the comprehensiveness and credibility of comprehensive evaluation.

2. *Principle of indexes measurability*. The meaning of indexes is clear, the data required for calculating indexes are easy to collect, and the calculation method is simple and easy to master.

3. *Principle of correlation between indexes and targets*. The realization of indexes must make a substantial contribution to the realization of the goal, and it is forbidden to choose indexes unrelated to the goal.

**Measurement indexes screening and weighting.** *1) Reliability test of measurement indexes*. Bartlett's spherical test and KMO (Kaiser-Meyer-Olkin) test are performed on the measurement index to determine whether the data of the measurement index meet the conditions of factor analysis. Result suggests that the KMO value of the measurement index after the initial screening was $0.771 > 0.5$, Sig. $= 0 < 0.05$, indicating that the measurement indicators after the initial screening meet the conditions of factor analysis.

*2) Factor principal component analysis*. Principal component analysis is performed on the initially determined measurement indexes, and the factor with a cumulative contribution rate of 85% is used as the final measurement indexes.

*3) Measurement indexes weight*. The expert assignment method is used to analyze the correlation between the measurement indexes and the evaluation object to provide data support for determining the weight of the measurement indexes. This paper uses Likert's 5-level scale method to assign values to the measurement indexes: "strong correlation" = 5, "high correlation" = 4, "general" = 3, "weak correlation" = 2, "uncorrelated" = 1. Using the coefficient of variation method to determine the weight of the measurement indexes, the steps are as follows:

Step 1: After preprocessing the original data, calculate the mean and standard deviation of each measurement indexes.

Step 2: Calculate the coefficient of variation of each measurement indexes based on the mean and standard deviation according to formula (1).

$$v_i = \frac{\partial_i}{\bar{x}_i} \tag{1}$$

Where $v_i$ is the coefficient of variation of the $i$-th measurement index, $\partial_i$ is the standard deviation of the $i$-th measurement indexes, and $\bar{x}_i$ is the average score of the $i$-th measurement indexes.

Step 3: Calculate the sum of coefficients of variation of each measurement index according to formula (2).

$$A = \sum_{i=1}^{n} v_i, \ i = 1, 2, \ldots, n \tag{2}$$

Step 4: Calculate the weight of each measurement index according to formula (3). The weight of each measurement index is equal to the ratio of the coefficient of variation of the

measurement index and the coefficient of variation.

$$w_i = \frac{v_i}{A} \tag{3}$$

According to the above methods, the indicators and weights for the collaborative efficiency of emergency information sharing in coastal cities were finally determined, as shown in Table 1.

*4) Method of obtaining original data of indexes.* As most of the measurement indexes are qualitative, it is difficult to obtain data through monitoring equipment, so the method of expert assignment value is adopted to obtain the original data, and five levels of quantity are adopted to assign values to each measurement indicator: good = 5, good = 4, fair = 3, poor = 2 and poor = 1.

## Method for measuring collaborative efficiency of emergency information sharing [29]

This paper used the coupling theory in physics to quantitatively evaluate the collaborative efficiency of emergency information sharing in coastal cities. Coupling theory advocates using the degree of coupling to measure the level of coordination among various factors within the system, which is similar to the coordination between various measurement indicators in the coastal city information sharing system. Therefore, this paper used the coupling degree in physics to evaluate the efficiency of emergency information sharing and collaboration in coastal cities.

**Calculate the cooperation efficiency of each subsystem.** The three dimensions in Table 1 were regarded as subsystems. Firstly, the contribution of each measurement index to the subsystem was determined, and Let the variable $U_i$(i = 1,2, . . ., n) be the sequence parameter for the measurement of the coordinated efficiency of emergency information sharing in coastal

**Table 1. Coastal city emergency information sharing collaborative efficiency measurement indexes.**

| Dimension | Impact factor |
|---|---|
| Construction level of emergency information sharing mechanism | Emergency information collection and processing$C_{11}$ (0.201) |
| | Emergency information report$C_{12}$ (0.170) |
| | Emergency information disclosure$C_{13}$ (0.181) |
| | Emergency information supervision and regulation$C_{14}$ (0.132) |
| | Emergency information security$C_{15}$ (0.135) |
| | Emergency information sharing incentives$C_{16}$ (0.181) |
| Emergency information sharing resource support capability | Emergency information technology and equipment$C_{21}$ (0.277) |
| | Completeness of policies and regulations$C_{22}$ (0.264) |
| | Uniformity of emergency information standards$C_{23}$ (0.281) |
| | Capital guarantee$C_{24}$ (0.178) |
| Synergistic driving force for emergency information sharing | Clear management function positioning$C_{31}$ (0.257) |
| | Emergency information service satisfaction$C_{32}$ (0.167) |
| | Barriers to emergency information sharing$C_{33}$ (0.310) |
| | Emergency management organization integration capabilities$C_{34}$ (0.266) |

cities, that was, the subsystem of "construction level of emergency information sharing mechanism", the subsystem of "support capacity of emergency information sharing resources" "Emergency Information Sharing Collaborative Driving Force" subsystem, then the emergency information sharing collaborative efficiency function is:

$$U_{ij} = \begin{cases} \dfrac{\left(X_{ij} - \beta_{ij}\right)}{\alpha_{ij} - \beta_{ij}}, & U_{ij} \text{ is positive efficiency} \\[2ex] \dfrac{\left(\alpha_{ij} - X_{ij}\right)}{\alpha_{ij} - \beta_{ij}}, & U_{ij} \text{ is negative efficiency} \end{cases}, \; i = 1, 2, \ldots, n; j = 1, 2, \ldots, m \quad (4)$$

In formula (4), $U_{ij}$ is the $j$-th measurement index of the $i$-th order parameter, and its value $X_{ij}$, $X_{ij}$ reflects the satisfaction of each measurement index in each subsystem to achieve the target efficiency, ranging from 0 to 1, approaching 0 is the most dissatisfied, approaching 1 is the most satisfactory; $\alpha_{ij}, \beta_{ij}$ is the upper and lower limits of the order parameters at the critical point of system stability.

The overall efficiency of each subsystem is calculated using the linear weighting method:

$$U_i = \sum_{j=1}^{m} w_{ij} U_{ij} \quad (5)$$

Among them, $w_{ij}$ is the weight of the j-th measurement index in the $i$-th subsystem, $U_i$ is the contribution of each subsystem to the order of the total system, and m is the number of measurement indexes in each subsystem.

**The total efficiency of emergency information sharing and coordination in coastal cities.** According to the coupling degree model in physics, the measurement function of the cooperative efficiency of the three subsystems is:

$$C = \sqrt[3]{(u_1 u_2 u_3)/(u_1 + u_2)(u_2 + u_3)(u_3 + u_2)} \quad (6)$$

In formula (6), the value of C is between 0 and 1. When C = 0, it indicates that the coordination efficiency between the subsystems is extremely small and is in an irrelevant state; when C = 1, it indicates high coordination efficiency between subsystems.

The total efficiency of formula (6) is of great significance for judging the efficiency intensity among the three subsystems, but the collaborative efficiency of formula (6) is difficult to reflect the contribution of all subsystems to the entire system, and relying solely on synergy efficiency to determine the magnitude of the synergy efficiency of the entire system may cause errors, because when the development level of the subsystem is low, its total synergy efficiency may be high. Therefore, we need to improve Eq (6) to reflect the true synergy efficiency of the subsystem to the entire system,

$$\begin{cases} F = p_1 u_1 + p_2 u_2 + p_3 u_3 \\ C = \sqrt[3]{(u_1 u_2 u_3)/(u_1 + u_2)(u_2 + u_3)(u_3 + u_2)} \\ H = \sqrt{C \cdot F} \end{cases} \quad (7)$$

Among them, F is the comprehensive development index, considering the difference of the important level contributed by each subsystem, p1, p2, p3 are the weighting coefficients, and the general value is 0.333; H is the modified synergy efficiency value, which combines the total synergy efficiency level C and development index F.

**Table 2. Emergency information sharing coordination efficiency level.**

| H | [0,0.4) | [0.4,0.6) | [0.6,0.8] | (0.8,1) |
|---|---------|-----------|-----------|---------|
| Efficiency class | low | medium | high | Extremely high |

In order to clarify the collaborative efficiency level of emergency information sharing in coastal cities, this paper refers to the hierarchical classification method proposed by Su Yi et al. (2018) [30] to divide the H value into 4 levels, as shown in Table 2.

## Empirical analysis

### Data acquisition

In this paper, 13 disaster events in coastal cities in China in the past 20 years are selected as samples, as shown in Table 3. The reason for choosing these samples is that these samples represent almost all the characteristics of coastal cities in China, and they are all major marine disasters. These sample data are relatively complete and have been collected on. Because most of the measurement indexes are qualitative indicators, it is difficult to obtain data through monitoring equipment, so the method of expert assignment is used to obtain the original data, and the method of 5-level quantity is used to assign values to each measurement indicator: "Good" = 5, "Better" = 4. "General" = 3, "Poor" = 2, "Bad" = 1. 45 experts were invited, including 6 maritime department management personnel, 10 emergency management personnel, 6 port management personnel, and 23 experts and scholars. Cronbach's α coefficient and Bartlett test were used to test the reliability and validity of the sample data. The results showed that the sample data assigned by experts can objectively reflect the attributes of 13 disaster events. The final score of each measurement index is counted according to the mode.

**Table 3. 13 disaster events in coastal cities of China in the past 20 years.**

| Year | Type of disaster | Time of occurrence | Disaster Covered City | Disaster situation |
|------|------------------|--------------------|-----------------------|--------------------|
| 2002 | Red tide | May 10 | Ningbo | Affected sea area 100km$^2$ |
| 2020 | typhoon | August 1 | Xiangshan | Wind power level 12, 278.8 thousand acres of affected farmland, 3625 people were killed and 3825 people were injured |
| 2020 | typhoon | September 3 | Lianjiang | Wind power level 10, 4.116 million mu of affected farmland, 134 deaths |
| 2020 | typhoon | August 24 | Putuo | Wind power level 12, 1.007 million mu, 18 people killed and 55 injured |
| 2003 | Red tide | May 11 | East of Ningbo Nantian Island | The affected sea area is about 2000km$^2$ |
| 2004 | Red tide | June 1–5 | The waters near the Yushan Islands | Affected sea area is 1000km$^2$ |
| 2006 | Red tide | June 7 | Xiangshan near shore | Affected sea area is 1000km$^2$ |
| 2001 | Heavy rain | August 9–11 | Zhanjiang, Maoming, Meizhou | 1.165 million people were affected and 3665 houses collapsed |
| 2010 | Red tide | June 9 | Songlan Mountain to Tantou Mountain | Affected sea area 1600km$^2$ |
| 2020 | Heavy rain | May 7 | Guangzhou | 109 houses collapsed, 256,800 mu of farmland was submerged, 32,166 people were affected, 6 died, and the direct economic loss was 543.8 million yuan |
| 2007 | Storm surge | October 11–12 | Hebei Province, Tianjin City, Shandong Province | Direct economic loss of 1.3 billion yuan, 200,000 people affected by the disaster |
| 2007 | Storm surge | March 4 | Weihai, Yantai, Weifang, Qingdao, Binzhou, Dongying | Direct economic loss of 1.927 billion yuan, 3 deaths, 7 missing |
| 2018 | Storm surge | September 19 | Zhuhai, Zhongshan, Jiangmen, Yangjiang, Zhanjiang, Maoming | 6.52 million people were affected, 26 people were killed and missing, 15,322 houses collapsed, and direct economic loss was 11.38 billion yuan |

**Table 4. Collaborative efficiency of emergency information sharing for 13 disaster events.**

| Serial number | Emergency information sharing mechanism disaster reduction level U1 | Emergency information sharing resource support capability U2 | Emergency information sharing collaborative promotion capability U3 | Comprehensive Development Index F | Overall system coordination efficiency C | Modified collaborative efficiency H |
|---|---|---|---|---|---|---|
| 1 | 0.078 | 0.156 | 0.172 | 0.135 | 0.090 | 0.110 |
| 2 | 0.312 | 0.417 | 0.192 | 0.307 | 0.219 | 0.259 |
| 3 | 0.089 | 0.332 | 0.467 | 0.296 | 0.244 | 0.269 |
| 4 | 0.267 | 0.231 | 0.189 | 0.229 | 0.165 | 0.194 |
| 5 | 0.478 | 0.329 | 0.172 | 0.326 | 0.222 | 0.269 |
| 6 | 0.648 | 0.099 | 0.219 | 0.322 | 0.173 | 0.236 |
| 7 | 0.389 | 0.098 | 0.319 | 0.269 | 0.195 | 0.229 |
| 8 | 0.178 | 0.091 | 0.319 | 0.196 | 0.158 | 0.176 |
| 9 | 0.419 | 0.315 | 0.091 | 0.275 | 0.150 | 0.203 |
| 10 | 0.538 | 0.458 | 0.510 | 0.502 | 0.504 | 0.503 |
| 11 | 0.610 | 0.309 | 0.698 | 0.539 | 0.573 | 0.556 |
| 12 | 0.289 | 0.319 | 0.107 | 0.238 | 0.140 | 0.183 |
| 13 | 0.427 | 0.710 | 0.300 | 0.479 | 0.389 | 0.431 |

According to the synergy efficiency measurement method in Section 3.2, the contribution level of each subsystem's order parameter, the comprehensive index F, the system's total synergy efficiency C, and the modified system efficiency H are calculated in sequence. Since the three subsystems are equally important, the values of p1, p2, and p3 are 0.333, and the calculation results are shown in Table 4.

## Results analysis and policy recommendations

**The overall efficiency of China's coastal cities' emergency information sharing is low.** The revised collaborative efficiency in Table 4 is classified according to the classification criteria in Table 2. The results show that the revised collaborative efficiency levels of Event 10, Event 11 and Event 13 are medium, and their revised collaborative efficiency values are 0.503, 0.556 and 0.431, respectively. The coordination efficiency of the remaining 11 disaster events is of low grade, with the highest revised coordination efficiency value of 0.269 and the lowest of 0.110, indicating that the collaborative efficiency of emergency information sharing in coastal cities in China is generally of low to medium grade.

**Although the contribution level of each subsystem has experienced twists and turns, it has generally increased.** The 13 events in Table 4 are arranged in chronological order. The calculation results show that, on the whole, the contribution levels of the subsystems U1 = emergency information sharing mechanism construction level, U2 = emergency information sharing resource support capability and U3 = emergency information sharing collaborative driving force are obviously increasing, which indicates that with the improvement of disaster prevention and mitigation capability of coastal cities in China, the collaborative capability of emergency information sharing is also increasing year by year.

**The contribution levels of the three subsystems are uneven.** The average contribution level of the U1 subsystem among the three subsystems is 0.363 higher than the average contribution level of the U2 and U3 subsystems, indicating that the construction level of the emergency information sharing mechanism has increased significantly and plays a key role in the total contribution. The contribution level is still low, with an average contribution rate of only 0.316.

Policy suggestion: The above analysis results showed that the overall low efficiency of emergency information sharing collaboration in coastal cities in China is due to the low level of emergency information sharing mechanism construction, weak emergency information sharing resource support capabilities, and insufficient emergency information sharing synergy, the following recommendations are made:

**Speed up the construction of emergency information sharing mechanism system in coastal cities.** Focus on strengthening emergency information collection and processing, emergency information sharing incentives, emergency information disclosure and reporting mechanism construction, improve emergency information supervision and regulation capabilities, and strengthen emergency information security guarantee measures;

**Enhancing the emergency shared resources guarantee capability of coastal cities of China.** Accelerating the formulation of emergency information standards, improving relevant information management policies and regulations, and improving emergency information processing technology and hardware and software equipment;

**Comprehensively accelerating the construction of a coordinated driving force for emergency information sharing in coastal cities.** Accelerating to reduce the barriers to emergency information sharing, clarifying emergency information management functions, accelerating the integration of institutions across departments and cities, and increasing emergency information service satisfaction.

## Conclusions

Aiming at the problem of emergency information sharing mechanism and collaborative efficiency measurement in the coordinated management of disaster prevention and mitigation in coastal cities of China, a theoretical framework and operation mechanism of emergency information sharing based on the collaborative theory were proposed, and It also proposed the measurement indexes system and measurement method of coastal city emergency information sharing collaborative efficiency around the construction of coastal city emergency information sharing mechanism, resource guarantee capability and emergency information sharing.

### Established the theoretical framework and operation mechanism of emergency information sharing in coastal cities

Introduce the theory of synergetic into the construction of emergency information sharing framework, by clarifying the operational relationship between the participants of emergency information sharing and information transmission, construct an emergency information sharing framework and operation mechanism suitable for the actual disaster prevention and reduction of coastal cities. Put forward an index system for measuring the efficiency of collaborative emergency information sharing in coastal cities Based on the innovation communication theory, existing research results and extensive interviews, around the three dimensions of emergency information sharing mechanism construction, resource support capabilities and emergency information sharing synergy, through the selection of indicators. I propose that I can effectively evaluate emergency information index system for sharing collaborative efficiency.

### Propose a method for measuring the collaborative efficiency of emergency information sharing in coastal cities based on the coupling degree model

With the help of the coupling theory in physics, through the evaluation of the contribution level of each subsystem and the modified coordination efficiency of the entire system, it is

found that the efficiency of emergency information sharing in coastal cities in China is on the rise, and the contribution rate of the level of emergency information sharing mechanism construction is higher than the resource guarantee capability. Synergy with emergency information sharing. But overall, the efficiency of China's coastal cities in emergency information sharing is still at a low to medium level.

To a certain extent, this paper enriches the research content of emergency management, but in order to avoid the difficulty of obtaining monitoring data, the method of expert assignment is used to obtain the data of measurement indicators, which to a certain extent affects the credibility of the evaluation results, which needs to be further studied.

## Acknowledgments

We wish to thank experts and journal editors who reviewed this article. We also wish to thank all scholars who provided references.

## Author Contributions

**Conceptualization:** Zou Jiang.

**Investigation:** Zou Jiang.

**Methodology:** Huang Xing.

**Supervision:** Huang Xing.

**Writing – original draft:** Huang Xing.

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
