## [Decision Letter · Decision Letter 0]

1 May 2022

PONE-D-21-09794Emergency Information Sharing Mechanism and Efficiency Measurement in Coastal Cities Based on the Perspective of Collaborative GovernancePLOS ONE

Dear Dr. Huang,

Thank you for submitting your manuscript to PLOS ONE; I sincerely apologize for the unusually delayed review timeframe. After careful consideration, we feel that it has merit but does not fully meet PLOS ONE’s publication criteria as it currently stands. Therefore, we invite you to submit a revised version of the manuscript that addresses the points raised during the review process.

Please note that we have only been able to secure a single reviewer to assess your manuscript. We are issuing a decision on your manuscript at this point to prevent further delays in the evaluation of your manuscript. Please be aware that the editor who handles your revised manuscript might find it necessary to invite additional reviewers to assess this work once the revised manuscript is submitted. However, we will aim to proceed on the basis of this single review if possible.

We look forward to receiving your revised manuscript.

Kind regards,

Emily Chenette

Editor in Chief

PLOS ONE

Journal Requirements:

3. Thank you for stating the following financial disclosure: "Initials of the authors who received each award Grant numbers awarded to each author"

4. Thank you for stating the following in the Acknowledgments Section of your manuscript: "This article is funded by the National Social Science Foundation of China (18XGL016), thanks to the review experts and editors of this article, and to the Chinese Ministry of Natural Resources and other government agencies that provide data support for this article. All data, models, and code generated or used during the study appear in the submitted article."

Please remove any funding-related text from the manuscript and let us know how you would like to update your Funding Statement. Currently, your Funding Statement reads as follows: "Initials of the authors who received each award Grant numbers awarded to each author"

6. Please include a copy of Table 3 which you refer to in your text on page 11.

Reviewers' comments:

Reviewer's Responses to Questions

**Comments to the Author**

1. Is the manuscript technically sound, and do the data support the conclusions?

Reviewer #1: Yes

2. Has the statistical analysis been performed appropriately and rigorously? 

Reviewer #1: Yes

3. Have the authors made all data underlying the findings in their manuscript fully available?

Reviewer #1: No

4. Is the manuscript presented in an intelligible fashion and written in standard English?

Reviewer #1: Yes

5. Review Comments to the Author

Reviewer #1: Overall Comments:

This paper constructs the framework and operation mechanism of emergency information sharing in Coastal Cities and proposes the evaluation index system of emergency information sharing collaborative efficiency. In general, the topic in this paper is interesting and fits well within the scope of this journal. However, there are still some details that need to be improved. Thus, the paper in its present form is not ready for publication. The following comments and feedback could hopefully assist the authors in future revisions.

Specific Comments:

1. Introduction: i) What are the emergency public emergencies? It is necessary to demonstrate the difference between coastal cities and inland cities. ii) How can the establishment of an institution (National Emergency Management Department) “completely” changed the pattern? iii) The lack of comparison between other countries' coastal cities in the same period, otherwise, it is not appropriate to directly say that China's low level. iv) Literature reviews should be classified based on research content. v) I don't think the existing research lacks research on the emergency mechanism of coastal cities, given that there are so many coastal cities in the world. Or what is special to the coastal city emergency management system.

2. Emergency Information Sharing Framework Based on Collaboration Theory: i) Are the four elements of the coastal city emergency information sharing framework original? The name of the “information participants” is not appropriate, and it contains too much content. ii) As previously written, “The emergency response department is mainly responsible for executing the emergency tasks and its responsibilities issued by the command center”. Information collection is not its responsibility. And how to conclude that “the emergency response department is not active, relying on the decision and instructions of the emergency command center, and the efficiency of information collection, disposal and feedback is low.”

3. Coastal city emergency information sharing collaborative efficiency measurement: Whether this Coastal city emergency information sharing collaborative efficiency measurement is common to coastal cities and inland cities?

4. Empirical analysis: “The overall efficiency of China's coastal cities' emergency information sharing is low”, This indicator is low in the world, and it is still relatively low compared to inland cities?

5. Editorial errors: i) “Basic information resources include five aspects: politics, economy, social development, resources, environment, and geographic information”, This statement is incorrect. ii) The words in the text box in Figure 2 should be left intact and a hyphen is required for line breaks. iii) In the “Policy suggestion”, at the end of the second suggestion is the fourth.

6. PLOS authors have the option to publish the peer review history of their article (what does this mean?). If published, this will include your full peer review and any attached files.

Reviewer #1: No

---

## [Author Response · Author response to Decision Letter 0]

15 Jun 2022

Thank the reviewer for providing the direction for the author to improve the manuscript. According to the reviewer's suggestions, the author has made the modifications and explanations to the manuscript:

1. Introduction: i) What are the emergency public emergencies? It is necessary to demonstrate the difference between coastal cities and inland cities. ii) How can the establishment of an institution (National Emergency Management Department) “completely” change the pattern? iii) The lack of comparison between other countries' coastal cities in the same period, otherwise, it is not appropriate to directly say that China's low level. iv) Literature reviews should be classified based on research content. v) I don't think the existing research lacks research on the emergency mechanism of coastal cities, given that there are so many coastal cities in the world. Or what is special to the coastal city emergency management system.

Author response:

i) Public emergencies refer to negative events that occur unexpectedly under the domination of certain natural factors, cause serious harm, loss or impact to the society and need to be dealt with immediately. At the beginning of this paper, the special features of coastal cities and their special factors for disasters are proposed.

ii) In this paper, the "completely changing model" is changed to emphasize the improvement of emergency management capabilities.

iii) The gap between China's coastal cities and foreign countries can be seen from the literature review by foreign scholars on the emergency management capabilities of coastal cities.

iv) The literature review is organized from the aspects of emergency information sharing collaborative research, emergency information sharing mechanism research and emergency collaborative governance.

v) Firstly, the research of this paper is based on the theory of collaborative governance. Secondly, this paper proposes a framework and operation mechanism for coastal cities' emergency information sharing based on the collaborative theory, and proposes an index system and measurement method that affects the collaborative efficiency of coastal cities' emergency information sharing. This paper provided support for disaster prevention and mitigation information sharing decision-making.

2. Emergency Information Sharing Framework Based on Collaboration Theory: i) Are the four elements of the coastal city emergency information sharing framework original? The name of the “information participants” is not appropriate, and it contains too much content. ii) As previously written, “The emergency response department is mainly responsible for executing the emergency tasks and its responsibilities issued by the command center”. Information collection is not its responsibility. And how to conclude that “the emergency response department is not active, relying on the decision and instructions of the emergency command center, and the efficiency of information collection, disposal and feedback is low.”

Author response:

i) The four elements of the coastal city emergency information sharing framework are original. Emergency management information involves the coordination of various government departments. The research in this paper includes all relevant departments as much as possible.

ii) The main responsibility of the emergency response department is indeed not information collection, but before it orders tasks on its own, it needs to adjust its own response strategy in time based on the information collected. The conclusions drawn in this paper are based on the relevant literature and research results.

3. Coastal city emergency information sharing collaborative efficiency measurement: Whether this Coastal city emergency information sharing collaborative efficiency measurement is common to coastal cities and inland cities?

Author response: The collaborative efficiency measure of emergency information sharing in coastal cities is an influencing factor refined based on the characteristics of coastal cities and the theory of collaborative governance. Therefore, it is general for coastal cities, but not suitable for inland cities.

4. Empirical analysis: “The overall efficiency of China's coastal cities' emergency information sharing is low”, This indicator is low in the world, and it is still relatively low compared to inland cities?

Author response: This metric is lower worldwide, not just within China.

5. Editorial errors: i) “Basic information resources include five aspects: politics, economy, social development, resources, environment, and geographic information”, This statement is incorrect. ii) The words in the text box in Figure 2 should be left intact and a hyphen is required for line breaks. iii) In the “Policy suggestion”, at the end of the second suggestion is the fourth.

Author response:

i) Resources and environment are not two words, but one word reflecting two aspects, and the author has corrected the English translation.

ii) The author has adjusted the formativeness of the graph.

iii) The author changed (4) to (3)

In addition to making revisions based on the above suggestions, we have also improved the grammatical structure of the manuscript. We hope that reviewers can make further suggestions for revisions. Thank reviewer again!

---

## [Decision Letter · Decision Letter 1]

8 Sep 2022

PONE-D-21-09794R1Influencing Factors and Efficiency Evaluation of Coastal Emergency Information SharingPLOS ONE

Dear Dr. Huang,

Thank you for submitting your revised manuscript to PLOS ONE. Your manuscript has been assessed by two reviewers: the original reviewer, who is positive about the revisions, and one new reviewer. Although reviewer 1 is positive about the revisions, reviewer 2 raises some additional concerns that should be addressed as a prerequisite to further consideration of this work. Therefore, we invite you to submit a revised version of the manuscript that addresses the points raised during the review process.

We look forward to receiving your revised manuscript.

Kind regards,

Emily Chenette

Editor in Chief

PLOS ONE

Journal Requirements:

Reviewers' comments:

Reviewer's Responses to Questions

**Comments to the Author**

1. If the authors have adequately addressed your comments raised in a previous round of review and you feel that this manuscript is now acceptable for publication, you may indicate that here to bypass the “Comments to the Author” section, enter your conflict of interest statement in the “Confidential to Editor” section, and submit your "Accept" recommendation.

Reviewer #1: All comments have been addressed

Reviewer #2: (No Response)

2. Is the manuscript technically sound, and do the data support the conclusions?

Reviewer #1: Yes

Reviewer #2: Partly

3. Has the statistical analysis been performed appropriately and rigorously? 

Reviewer #1: N/A

Reviewer #2: Yes

4. Have the authors made all data underlying the findings in their manuscript fully available?

Reviewer #1: No

Reviewer #2: Yes

5. Is the manuscript presented in an intelligible fashion and written in standard English?

Reviewer #1: Yes

Reviewer #2: No

6. Review Comments to the Author

Reviewer #1: In this revision, all the reviewer's comments have been addressed. The reviewer does not have further comments.

Reviewer #2: This paper “Influencing Factors and Efficiency Evaluation of Coastal Emergency Information Sharing” constructs the framework of emergency information sharing in coastal cities and adopts coupling degree model to evaluate emergency information sharing collaborative efficiency. The perspective of this study is interesting, and the results provide support for improving the disaster prevention and mitigation capacity of coastal cities.

However, the authors provided relatively few conceptual details on the rationality of the selection of the evaluation indicators and the originality of the approach. This is probably the main flaw of this paper, preventing the publication of this article as it is. The suggestions for specific improvement are as follows.

Major reviews:

The first one regards Introduction and Discussion. The manuscript fails to address how the findings relate to previous research in this area. In addition, 22 of 28 references regard Chinese case studies. It seems that the proposed study regards only this specific geographic position. The authors should rewrite their Introduction and add Discussion to reference more related literature, especially recently published international work. The authors should also elaborate better the originality and novelty of their study, commenting a bit more on the representativeness and the generalizability of the study to broader socioeconomic contexts.

The second one regards Section 3: Coastal city emergency information sharing collaborative efficiency measurement. This study requires more conceptual details on the choice of the indicators and methods. The authors should include more information that clarifies and justifies their choice of methods. In addition, it is better to elaborate more on the meaning of the indicators, the gap and implication for the evaluation system taking the previous example for better understanding.

The third one regards Section 4: Empirical analysis. 13 disaster events that occurred in coastal cities in China in the past 20 years are selected as samples. On what basis the authors selected those events as samples and representativeness of these samples should be more elaborate. Additionally, it was mentioned in the paper that contributions of U1, U2 and U3 subsystem had risen steadily. However, the presented data result in Table 4 does not fully support the conclusion. The authors should check carefully and confirm that the conclusions are consistent with the research results, and the recommendations are corresponding to the conclusions.

Other minor reviews:

i) The statement in paragraph 1 of Section 1 and Section 2.3 should be justified by citations.

ii) Emergency information participants and emergency information resources are in a parallel relationship. The presentation of the relationship between them in Figure 1 seems to be an inclusive relationship, which is obviously inappropriate.

iii) Figure 3 is mentioned in Section 2.3 (1). However, there is no Figure 3 in the revised manuscript.

iv) Is it emergency efficiency or collaborative efficiency in the title of Section 3?

v) Please add the year information for the time of occurrence in Table 3.

vi) There are some typos and grammar mistakes in this manuscript. Please carefully check and correct these mistakes. Furthermore, moderate revisions are requested in order to polish the language usage (longer sentences and technicalities).

vii) Please make sure all dash, space, a hyphen, en dash, and capital words would be appropriate throughout the manuscript.

viii) Please ensure that the font size and capital letters are consistent in the table and figure.

All in all, I have the impression these revisions will improve substantially both quality and readability of the present manuscript. Thank you.

7. PLOS authors have the option to publish the peer review history of their article (what does this mean?). If published, this will include your full peer review and any attached files.

Reviewer #1: No

Reviewer #2: No

---

## [Author Response · Author response to Decision Letter 1]

14 Sep 2022

Reviewer #2: 

Question 1: Introduction and Discussion: The manuscript fails to address how the findings relate to previous research in this area. In addition, 22 of 28 references regard Chinese case studies. It seems that the proposed study regards only this specific geographic position. The authors should rewrite their “Introduction” and add “Discussion” to reference more related literature, especially recently published international work. The authors should also elaborate better the originality and novelty of their study, commenting a bit more on the representativeness and the generalizability of the study to broader socioeconomic contexts.

Author response: Author adds the following contents in the part of introduction to increase to increase foreign literatures : Carminati B et al(2016)[14] presented an emergency information sharing framework able to deal with both specified and unspecified emergencies; Dantas A A et al(2007)[15] presented an information-sharing framework for road organizations, on the basis of a study of response and recovery activities, information needs were identified and a geographic information system-based information-sharing framework was created.

Author adds the following contents in the part of introduction to increase to find relate to previous research in this area and to explain this innovation: The relationship between this paper and the previous research lies in: First, the index system of the existing research is optimized, and an effective evaluation index system is put forward from three dimensions, included the mechanism construction level of emergency information sharing, the resource support capability and the collaborative driving force. Secondly, the coupling theory in physics is used for reference to quantitatively evaluate the coordination efficiency of emergency information sharing in coastal cities. Thirdly, referring to the existing emergency information organization structure, the paper puts forward the emergency information sharing framework of China's coastal cities based on collaboration theory. The difference of this paper is that it proposes a coastal city emergency information sharing framework and operation mechanism based on the synergy theory, and proposes an index system and measurement method that affect the coastal city emergency information sharing collaboration efficiency.

Author also adds some references, as follows: 

[14] Carminati B, Ferrari E, Guglielmi M, 2016, Detection of Unspecified Emergencies for Controlled Information Sharing. IEEE Transactions on Dependable and Secure Computing, 13(6):630-643.

[15] Dantas A A, Sevile E and Gohil D,2007, Information sharing during emergency response and recovery - A framework for road organizations, Transportation Research Record, 2022,21-28.

[24] Kwon T. H, Zmud R. W., 1987, Unifying the fragmented models of information systems implementation, critical Issues in Information Systems Research. New York: John Wiley: 135 -142.

[25] Varun Grover, Martin D. Goslar, 1993, The initiation, adoption, and implementation of telecommunications technologies in U.S. organizations. Journal of Management Information Systems, 10(1): 141-163.

Question 2: Section 3: Coastal city emergency information sharing collaborative efficiency measurement. This study requires more conceptual details on the choice of the indicators and methods. The authors should include more information that clarifies and justifies their choice of methods. In addition, it is better to elaborate more on the meaning of the indicators, the gap and implication for the evaluation system taking the previous example for better understanding.

Author response: Author adds some contents in the part of section 3, as follows:

(1) Principles of index selection

1)Systematization principle

Indexes can reflect both direct and indirect effects, and ensure the comprehensiveness and credibility of comprehensive evaluation.

2) Principle of index measurability

The meaning of indexes is clear, the data required for calculating indexes are easy to collect, and the calculation method is simple and easy to master.

3) Principle of correlation between indexes and targets

The realization of indexes must make a substantial contribution to the realization of the goal, and it is forbidden to choose indexes unrelated to the goal.

Author also adds some contents about obtaining original data of indexes, as follows:

4) Method of obtaining original data of indexes 

As most of the measurement indexes are qualitative, it is difficult to obtain data through monitoring equipment, so the method of expert assignment value is adopted to obtain the original data, and five levels of quantity are adopted to assign values to each measurement indicator: good =5, good =4, fair =3, poor =2 and poor =1.

Question 3: Section 4: Empirical analysis. 13 disaster events that occurred in coastal cities in China in the past 20 years are selected as samples. On what basis the authors selected those events as samples and representativeness of these samples should be more elaborate. Additionally, it was mentioned in the paper that contributions of U1, U2 and U3 subsystem had risen steadily. However, the presented data result in Table 4 does not fully support the conclusion. The authors should check carefully and confirm that the conclusions are consistent with the research results, and the recommendations are corresponding to the conclusions.

Author response: Author adds some contents in the part “Empirical analysis”, as follows:

The reason for choosing these samples is that these samples represent almost all the characteristics of coastal cities in China, and they are all major marine disasters. These sample data are relatively complete and have been collected on.

The author modified some contents in order to keep Table 4 consistent with the conclusion, as follows:

（2）Although the contribution level of each subsystem has experienced twists and turns, it has generally increased. 

The 13 events in Table 4 are arranged in chronological order. The calculation results show that, on the whole, the contribution levels of the subsystems U1= emergency information sharing mechanism construction level, U2= emergency information sharing resource support capability and U3= emergency information sharing collaborative driving force are obviously increasing, which indicates that with the improvement of disaster prevention and mitigation capability of coastal cities in China, the collaborative capability of emergency information sharing is also increasing year by year.

Other minor reviews:

i) The statement in paragraph 1 of Section 1 and Section 2.3 should be justified by citations.

Author response: The author has cited the literature according to the expert opinion.

iii) Figure 3 is mentioned in Section 2.3 (1). However, there is no Figure 3 in the revised manuscript.

Author response: The author has modified Figure 3 to Figure 2.

iv) Is it emergency efficiency or collaborative efficiency in the title of Section 3?

Author response: It is emergency collaborative efficiency. Author had modified the title.

v) Please add the year information for the time of occurrence in Table 3.

Author response: Author had increased the year in Table 3.

vi) There are some typos and grammar mistakes in this manuscript. Please carefully check and correct these mistakes. Furthermore, moderate revisions are requested in order to polish the language usage (longer sentences and technicalities).

Author response: Author had edited some typos and grammar mistakes to this manuscript again.

vii) Please make sure all dash, space, a hyphen, dash, and capital words would be appropriate throughout the manuscript.

Author response: The author proofread the whole manuscript again.

viii) Please ensure that the font size and capital letters are consistent in the table and figure.

Author response: Author proofread to the font size and capital letters in the table and figure.

Sincerely thank editors and peer experts for their sugge

---

## [Decision Letter · Decision Letter 2]

26 Sep 2022

PONE-D-21-09794R2Influencing Factors and Efficiency Evaluation of Coastal Emergency Information SharingPLOS ONE

Dear Dr. Huang,

Thank you for submitting your manuscript to PLOS ONE. After careful consideration, we feel that it has merit but does not fully meet PLOS ONE’s publication criteria as it currently stands. Therefore, we invite you to submit a revised version of the manuscript that addresses the points raised during the review process.

 Although the authors addressed most of the points raised by the reviewers there are some minor grammatical errors which need to be revised. You will need to correct all the errors raised by the reviewer 2, which can be found in the attachment.

We look forward to receiving your revised manuscript.

Kind regards,

Md Asaduzzaman, Ph.D.

Academic Editor

PLOS ONE

Journal Requirements:

Additional Editor Comments:

Although reviewer 1 has accepted the current version reviewer 2 raised some crucial grammatical mistakes and textual errors. The reviewer also suggested elaborating by providing a detailed discussion of the reasons for choosing some existing results in the proposed index for measuring the efficiency of emergency information sharing collaboration. Therefore, we would like to invite you to submit a revised version of the manuscript by correcting those errors and issues raised and also a detailed response to reviewer 2.

Reviewers' comments:

Reviewer's Responses to Questions

**Comments to the Author**

1. If the authors have adequately addressed your comments raised in a previous round of review and you feel that this manuscript is now acceptable for publication, you may indicate that here to bypass the “Comments to the Author” section, enter your conflict of interest statement in the “Confidential to Editor” section, and submit your "Accept" recommendation.

Reviewer #1: All comments have been addressed

Reviewer #2: (No Response)

2. Is the manuscript technically sound, and do the data support the conclusions?

Reviewer #1: Yes

Reviewer #2: Partly

3. Has the statistical analysis been performed appropriately and rigorously? 

Reviewer #1: N/A

Reviewer #2: Yes

4. Have the authors made all data underlying the findings in their manuscript fully available?

Reviewer #1: No

Reviewer #2: Yes

5. Is the manuscript presented in an intelligible fashion and written in standard English?

Reviewer #1: Yes

Reviewer #2: No

6. Review Comments to the Author

Reviewer #1: All my comments have been addressed. This reviewer does not have further comments. This paper is recommended to be published.

Reviewer #2: Comments and Suggestions for Authors：

The authors have revised the manuscript according to reviewer’s comments, and the quality of the manuscript has been improved a lot. However, there are still minor problems raised by the reviewer that have not been solved.

The first one regards language usage and grammar. I think it is necessary to employ a professional scientific editor of language to thoroughly check this manuscript for language usage, spelling, and grammar. There are still some mistakes in this revised version, which authors haven’t corrected. Some of the mistakes are marked in the attachment named Comments on the Revised Manuscript.

The second one regards references. References should not be cited on the title. The statement in Section 2.3 and Section 3.2 should be justified by citations. Instead of adding references to the title, the author should add references to the exact statement to demonstrate the rationality of the statement.

Please ensure that the capital letters are consistent in the table and figure.

7. PLOS authors have the option to publish the peer review history of their article (what does this mean?). If published, this will include your full peer review and any attached files.

Reviewer #1: No

Reviewer #2: No

---

## [Author Response · Author response to Decision Letter 2]

28 Sep 2022

Response to Reviewers

Journal Requirements:

Author response: Author reviewed the reference list to ensure that it is complete and correct again.

Additional Editor Comments:

Although reviewer 1 has accepted the current version reviewer 2 raised some crucial grammatical mistakes and textual errors. The reviewer also suggested elaborating by providing a detailed discussion of the reasons for choosing some existing results in the proposed index for measuring the efficiency of emergency information sharing collaboration. Therefore, we would like to invite you to submit a revised version of the manuscript by correcting those errors and issues raised and also a detailed response to reviewer 2.

Author response: Author invited scholars from English-speaking countries to update the grammatical errors and text errors of this manuscript again. The revised contents in the manuscript are marked in red.

Author added some contents in 3.1 section to provide a detailed discussion of the reasons for choosing some existing results, as followed:

These indexes include 3 primary indexes and 9 secondary indexes. The first fist-level index is the construction level of emergency information sharing mechanism, which has four secondary indexes, including emergency information reporting, emergency information security, and emergency information sharing incentives. The second first-level index is the resource support capability of emergency information sharing, which has two second-level indexes, including the unity of emergency information standards, and financial support. The third first-level index is the collaborative driving force of emergency information sharing, and it has three second-level indexes, including clear management functions, obstacles to emergency information sharing, and the integration ability of emergency management agencies. These indexes can scientifically evaluate the collaborative efficiency of information sharing in coastal cities.

---

## [Decision Letter · Decision Letter 3]

11 Oct 2022

Research on Measurement Indexes and Evaluation for the Collaborative Efficiency of Emergency Information Sharing in Coastal Cities of China

PONE-D-21-09794R3

Dear Dr. Huang,

We’re pleased to inform you that your manuscript has been judged scientifically suitable for publication and will be formally accepted for publication once it meets all outstanding technical requirements.

Kind regards,

Md Asaduzzaman, Ph.D.

Academic Editor

PLOS ONE

Reviewers' comments:

Reviewer's Responses to Questions

**Comments to the Author**

1. If the authors have adequately addressed your comments raised in a previous round of review and you feel that this manuscript is now acceptable for publication, you may indicate that here to bypass the “Comments to the Author” section, enter your conflict of interest statement in the “Confidential to Editor” section, and submit your "Accept" recommendation.

Reviewer #1: All comments have been addressed

Reviewer #2: (No Response)

2. Is the manuscript technically sound, and do the data support the conclusions?

Reviewer #1: Yes

Reviewer #2: Yes

3. Has the statistical analysis been performed appropriately and rigorously? 

Reviewer #1: N/A

Reviewer #2: Yes

4. Have the authors made all data underlying the findings in their manuscript fully available?

Reviewer #1: No

Reviewer #2: Yes

5. Is the manuscript presented in an intelligible fashion and written in standard English?

Reviewer #1: Yes

Reviewer #2: Yes

6. Review Comments to the Author

Reviewer #1: This reviewer does not have further comments, and recommends this manuscript to be published in this journal.

Reviewer #2: The authors have revised the manuscript according to reviewer’s comments, and the quality of the manuscript has been improved a lot. However, the authors must delete the references in the title of Section 2.3 and Section 3.2. References should not be cited on the title. Instead of adding references to the title, the author should add references to the exact statement to demonstrate the rationality of the statement.

7. PLOS authors have the option to publish the peer review history of their article (what does this mean?). If published, this will include your full peer review and any attached files.

Reviewer #1: No

Reviewer #2: No

---

## [Editor Report · Acceptance letter]

17 Nov 2022

PONE-D-21-09794R3 

Research on Measurement Indexes and Evaluation for the Collaborative Efficiency of Emergency Information Sharing in Coastal Cities of China 

Dear Dr. Xing:

I'm pleased to inform you that your manuscript has been deemed suitable for publication in PLOS ONE. Congratulations! Your manuscript is now with our production department. 

Kind regards, 

on behalf of

Dr. Md Asaduzzaman 

Academic Editor

PLOS ONE